# Clinical Application of Pectoralis Nerve Block II for Flap Dissection-Related Pain Control after Robot-Assisted Transaxillary Thyroidectomy: A Preliminary Retrospective Cohort Study

**DOI:** 10.3390/cancers14174097

**Published:** 2022-08-24

**Authors:** Min Suk Chae, Youngkyung Park, Jung-Woo Shim, Sang Hyun Hong, Joonseon Park, Il Ku Kang, Ja Seong Bae, Jeong Soo Kim, Kwangsoon Kim

**Affiliations:** 1Department of Anesthesiology and Pain Medicine, Seoul St. Mary’s Hospital, College of Medicine, The Catholic University of Korea, Seoul 06591, Korea; 2Department of Surgery, College of Medicine, The Catholic University of Korea, Seoul 06591, Korea

**Keywords:** PECS II block, robotic surgery, transaxillary, thyroidectomy, visual analogue scale

## Abstract

**Simple Summary:**

The main findings of the study are that pectoralis nerve block II (PECS II) may be a valuable analgesic option that alleviates flap dissection pain and stress during a robot-assisted transaxillary thyroidectomy (RATT) and reduces opioid consumption in the early recovery phase. Patients who received PECS II experienced a more comfortable recovery and required fewer painkillers. PECS II may serve as a valuable new pain relief modality in addition to the multimodal analgesic strategy for patients undergoing RATT. Although we have yet to investigate the optimal block duration, regions of sensory loss, optimal technique, and possible complications, our preliminary study suggests that PECS II reduces flap dissection pain and thus promotes recovery. Appropriate analgesia during RATT remains challenging, but is a key issue for postoperative recovery. A further prospective investigation is required to validate our results and establish the optimal pain control regimen for patients undergoing RATT.

**Abstract:**

Few studies have examined the clinical utility of ultrasonography-guided pectoralis nerve block II (PECS II) during wide flap dissection of a robot-assisted transaxillary thyroidectomy (RATT). We assessed the ability of PECS II to reduce postoperative pain. We retrospectively reviewed 62 patients who underwent elective RATT from December 2021 to April 2022 at Seoul St. Mary’s Hospital (Seoul, Korea). The patients were divided into a block group (n = 28, 50.9%) and no-block group (n = 27, 49.1%). Pain was measured using a visual analog scale (VAS) at 4, 10, 20, 25, 35, and 45 h after surgery, and the requirements for rescue painkillers in the post-anesthesia care unit and ward were recorded. The VAS scores did not differ significantly between the two groups at 4 h postoperatively. The block group had significantly lower VAS scores at 10 and 25 h (*p* = 0.017 and *p* = 0.034, respectively). The block group required fewer painkillers in the post-anesthesia care unit than the no-block group, although the difference was not statistically significant in the ward. PECS II may serve as a new pain relief modality and valuable addition to the current multimodal analgesic strategy for patients undergoing RATT.

## 1. Introduction

Surgical techniques for thyroid diseases have continuously evolved; prognosis and safety have been improved, and scarring, hospitalization times, and recurrence rates have been reduced. Modifications of operative methods have increased efficacy and reduced invasiveness; open incision was replaced by a videoscope-assisted technique [1,2,3]. However, thyroidectomy requires inflexible endoscopic devices guided by two-dimensional camera images, rendering it difficult to visualize the surgical field and manipulate the instrumentation. Surgical dissection approaches include anterior chest, breast, and transaxillary; gas insufflation and mechanical retractors are required [2,4,5]. The da Vinci robotic system was developed to improve both operative maneuverability (through multi-articulated instruments) and the surgical view (via a three-dimensional camera). Although the system has many advantages, skin incision, wide flap dissection, and pneumatic/mechanical retraction remain essential, but cause postoperative pain and slower recovery [6].

Regional thoracic anesthesia/analgesia was typically used for neuraxial epidural or paravertebral block. However, developments in ultrasound guidance led to new myofascial plane block techniques that have greatly aided many surgeries [7]. Previous studies found that pectoralis fascial blocks were easy to establish; local anesthetics are injected between two adjacent myofascial layers under ultrasound guidance, providing the surgeon with a clear image [7,8,9,10,11]. Analgesic efficacy has been validated during modified radical mastectomy, breast augmentation, and mastectomy with lymph node dissection. All of these surgeries require wide and painful dissection of chest wall structures [11,12]. Martsiniv et al. compared a pectoralis nerve block II (PECS II) to paravertebral block during breast cancer surgery. The pain outcomes were comparable between the groups, but there were fewer complications in the PECS II group [13]. Advantages include no risk of the sympathectomy occasionally associated with epidural and paravertebral blocks, and full coverage of the pectoralis, long thoracic, and thoracodorsal nerves. PECS II is thus valuable during extensive surgery involving axillary dissection [7,10,11].

To the best of our knowledge, no study has examined the clinical utility of ultrasonography-guided upper thoracic wall nerve block during wide flap dissection for a robot-assisted transaxillary thyroidectomy (RATT). We investigated whether this block can reduce postoperative pain.

## 2. Materials and Methods

### 2.1. Ethical Considerations

This preliminary, retrospective cohort study was conducted in accordance with the Declaration of Helsinki (2013) and approved by the institutional review board of Seoul St. Mary’s Hospital, Catholic University of Korea (Seoul, Korea) (approval number: KC22RISI0473); the board waived the requirement for informed consent given the retrospective nature of the work.

### 2.2. Study Population

We retrospectively reviewed 62 patients (aged ≥ 18 years) who underwent elective RATT (including lobectomy or total thyroidectomy) from December 2021 to April 2022 at Seoul St. Mary’s Hospital. The exclusion criteria were American Society of Anesthesiologists physical status III or IV (n = 3), conversion to non-robotic surgery (n = 2), a history of anticoagulation treatment (n = 1), and reoperation because of bleeding (n = 1). Ultimately, 55 patients were included, and divided into block (n = 28, 50.9%) and no-block (n = 27, 49.1%) groups. Based on the order of the surgical schedule, the patients with an odd-numbered position in the surgical schedule were assigned to the block group, whereas those with an even-numbered position were assigned to the no block group. A flow diagram of patient recruitment is shown in Figure 1.

### 2.3. RATT Proccedure

RATT involves three stages. First, flap dissection creates a working space from the axilla to the thyroid gland; the flap is lifted via external retraction. The second stage involves docking (positioning of the robotic arms) and the final (console) stage involves operation of the system by the surgeon. In this study, all procedures were performed by a single surgeon (K.K.) using the da Vinci single-port robotic system. The details of flap dissection have been described elsewhere [6]. The pectoralis major muscle was subjected to subcutaneous flap dissection via a skin incision in the axilla. A subplatysmal flap was lifted upward to expose the sternocleidomastoid muscle, the bifurcation of which was split into the sternal and clavicular heads to identify the strap muscles that surround the thyroid glands. Dissection continued under the strap muscles to expose the thyroid gland, and the flap was then raised (using an external retractor) to maintain the working space. The extent of flap dissection is shown in Figure 2. After the first stage (flap dissection), the docking and console stages were performed as recommended for RATT [6].

### 2.4. General Anesthesia

Propofol, rocuronium, and fentanyl were used to induce general anesthesia, and desflurane (in an air/oxygen mixture) and remifentanil were applied to maintain anesthesia. The monitoring modalities included electrocardiography, pulse oximetry, noninvasive blood pressure measurement, bispectral index assessment, and measurement of end-tidal carbon dioxide and esophageal temperature. At the end of surgery, a single dose (1 g) of acetaminophen (PROFA; Dai Han Pharm. Co. Ltd., Seoul, Korea) was injected into all patients for pre-emptive analgesia. The neuromuscular blockade was reversed by sugammadex under ventilation with 100% oxygen in the post-anesthesia care unit (PACU). In cases of moderate-to-severe postoperative pain (visual analog scale [VAS] score > 6 cm; 0 cm, no pain; 10 cm, worst possible pain), rescue analgesics were infused intravenously based on the discretion of the attending physicians: 50 μg of fentanyl in the PACU, and 25 mg of diclofenac (a non-steroidal anti-inflammatory drug [NSAID]) in the ward (as the first option; 25 mg of pethidine as the second option).

### 2.5. Ultrasonography-Guided PECS II

All pectoral nerve blocks were established by one highly experienced anesthesiologist (M.S.C.), and informed consent was obtained from each patient a day before surgery. After induction of general anesthesia with the patient supine, the left or right infraclavicular and axillary areas (depending on the dissection side) were cleaned with chlorhexidine, and an ultrasound probe was placed obliquely over the second and third ribs below the lateral one-third of the clavicle (i.e., on the anterior axillary line; Figure 3). After identification of the anatomical structures, an ultrasonography-guided block was induced via a medial in-plane approach using a 21-G echogenic needle (Echoplex; Vygon, Paris, France). The block was created in a deep (first)-to-superficial (second) order to avoid air bubbles (which reduce ultrasound image quality). The needle was advanced along a superior-medial-to-inferior-lateral passage to the tissue plane between the pectoralis minor and serratus anterior muscles, and 20 mL ropivacaine (0.375% *w/v*) was injected at the level of the third rib. The anesthetic spread around the axilla, and the needle was withdrawn to the point in the plane between the pectoralis major and minor muscles. A second injection of 20 mL ropivacaine (0.375% *w/v*) was then delivered (PECS II) [14,15]. Ultrasonography showed that the local anesthetics infiltrated the plane between the pectoralis major/minor and serratus anterior muscles, and there were no complications (such as arterial injury) (Figure 4).

### 2.6. Pain Outcomes

The VAS pain scores were recorded at 4, 10, 20, 25, 35, and 45 h after surgery, and the rescue painkiller requirements in the PACU and ward were logged.

### 2.7. Clinical Variables

We recorded patient age and gender, extent of the operation, pathology, and thyroid tumor features (size, multiplicity, minimal extrathyroidal extension status, lymph node involvement, and tumor, node, metastasis [TNM] stage).

### 2.8. Statistical Analysis

The normality of continuous variables was evaluated using the Shapiro–Wilk test. Continuous variables are reported as mean ± standard deviation and group differences were assessed using Student’s t-test or the Mann–Whitney U test. The paired t-test or Wilcoxon signed-rank test was used to compare VAS scores against those obtained at 4 h postoperatively. Categorical variables are reported as frequencies (percentages) and group differences were assessed using the Pearson chi-squared test or Fisher’s exact test. A two-sided *p*-value < 0.05 was considered statistically significant. All statistical analyses were performed with IBM SPSS Statistics for Windows (ver. 24.0; IBM Corp., Armonk, NY, USA).

## 3. Results

### 3.1. Baseline Clinicopathological Characteristics

Table 1 lists the clinicopathological characteristics of all patients. The mean age was 41.5 ± 12.4 (range: 18–69) years, and most of the patients (n = 52, 95.5%) were female. Forty-eight (87.3%) patients underwent lobectomy and seven (12.7%) underwent total thyroidectomy. The most common pathological diagnosis was papillary thyroid carcinoma (n = 45; 81.8%). The mean tumor size was 1.1 ± 1.0 (range: 0.3–5.3) cm. Thyroiditis (revealed by pathological review) was present in 25 (45.5%) patients with cancer, and in 46 (83.6%) patients overall. Most patients had TNM stage I cancer (n = 44, 95.7%). 

The clinicopathological characteristics of the block and no-block groups are summarized in Table 2. Age, sex, extent of the operation, pathology, tumor size, and TNM stage showed no significant group differences.

### 3.2. Postoperative Pain in the Block and No-Block Groups

Table 3 and Figure 5 show the postoperative pain data. There was no significant group difference in VAS score after 4 h (4.2 ± 2.0 vs. 4.4 ± 2.1, *p* = 0.628). The block group had significantly lower VAS scores at 10 and 25 h (*p* = 0.017 and *p* = 0.034, respectively). However, no significant group difference was observed after 35 h. The block group experienced an early (within 1 d of surgery; from 4 to 10 h) reduction in pain but the no-block group reported a reduction only after postoperative day 1. In terms of painkiller usage, the block group required fewer rescue analgesics in the PACU than the no-block group, although the difference was not statistically significant in the ward.

## 4. Discussion

The main finding of this study was that PECS II may be a valuable analgesic option for reducing flap dissection pain and stress, as well as opioid consumption during early recovery after RATT. PECS II may provide early pain relief; patients who received this block recovered more comfortably and require fewer painkillers.

RATT is an emerging, safe, and feasible technique providing outcomes comparable to, or better than, those of conventional open or endoscopic surgeries [16]. However, RATT requires a long and deep skin incision in the axillary crease, followed by wide and extensive dissection of a subcutaneous flap (using electrocautery) over the pectoralis muscles to the midline of the anterior neck, where the sternocleidomastoid muscle is directly identified. An external retractor is used to maintain the working space as the robot approaches the pathological thyroid site. This may cause mechanical strain in areas of dissection around tissues [6]. Between 35% and 65% of patients experience moderate-to-severe discomfort in the anterior chest area when the subplatysmal skin flap route is used, for at least 2 days after surgery [2,4,5]. Many surgeons have tested other surgical approaches, including gasless, bilateral axillo-breast, facelift (retro-auricular), and transoral approaches, but all have limitations such as the need for a larger work space (creating a huge wound), complicated techniques that increase the learning burden, and frequent collision among three multi-jointed instruments and the wrist camera (which reduces the maneuverability of RATT) [17,18,19,20]. Hong et al. reported that a single injection (1 g) of paracetamol (acetaminophen) provided greater postoperative pain relief than a placebo, but other studies found that paracetamol alone was not analgesic even after minor surgery [21,22,23].

Current pain control strategies are based on multimodal approaches, such as the enhanced recovery after surgery (ERAS) protocol; regional analgesic blocks are the cornerstones of such regimens [9]. During laparoscopic surgery, a transversus abdominis plane block-based pain-control regimen improved patient self-reported early postoperative recovery and reduced the pain score, rescue analgesic use, and nausea/vomiting [24]. During robot-assisted surgery, rectus sheath block therapy reduced postoperative pain both when resting and coughing [25]. From a safety perspective, regional analgesic block may moderately reduce pain and the opioid requirement, particularly in clinically vulnerable patients contraindicated for adjuvant painkillers such as nonsteroidal anti-inflammatory drugs (NSAIDs) or acetaminophen [26].

We found that PECS II can provide sufficient analgesia during RATT flap dissection, possibly by blocking the lateral cutaneous branches of the intercostal nerves (at approximately T2–T6, i.e., the long thoracic and thoracodorsal nerves) via analgesia of the upper anterolateral chest wall. PECS II (which includes PECS I) was induced by a second injection lateral to the PECS I injection point (between the pectoralis major and minor muscles) in the plane between the pectoralis minor and serratus anterior muscles. This blocked the upper intercostal nerves at the level of the third rib [27]. During various upper anterior trunk surgeries, PECS II has gradually emerged as a practical and efficacious alternative to central neuraxial block techniques (such as spinal and epidural methods) because it provides a relatively wide analgesic effect with fewer fatal complications [7,11,15,24,27,28]. PECS II was more helpful for patients undergoing extensive breast surgeries (mastectomy and resection of cancers in the pectoralis muscles, serratus anterior muscle, and axilla) than simple mass excision; the block reduced pain and morphine consumption more than systemic analgesic infusion, and was not inferior to neuraxial paravertebral block [11]. During cardiothoracic surgeries, wherein incisions involve the anterolateral chest wall, facial plane blocks (including PECS II) may reduce pain. Facial plane block dramatically improved minimally invasive surgeries, such as robot-assisted surgery [27,28]. We also found that pain in the early postoperative period (10–25 h) and painkiller consumption were lower in patients receiving PECS II, indicating that it aids recovery after RATT. It is not clear why PECS II was efficacious for analgesia; a cadaveric study is required. However, the anatomical region covered by the block is compromised during RATT.

Although rare, potential complications of a PECS II include infection, thoracoacromial artery injury and hematoma, pneumothorax, iatrogenic intravascular injection, and local anesthetic toxicity [8]. The latter complication can be fatal because it can trigger seizures or arrhythmia. Transversus plane blocks in patients with cardiac or renal dysfunction, those who received multiple injections, and healthy volunteers were associated with very large increases in plasma levels of local anesthetics [29]. Careless block maneuvers and a lack of sterility may be associated with site-related complications, pneumothorax, vascular injury, and infection [10]. We encountered none of these complications because a very experienced anesthesiologist created all of the blocks under ultrasound guidance, and there were no overdoses or repeat attempts at local anesthetic infusion.

This study had some limitations. First, as it was preliminary and retrospective, selection bias may have been present. Second, given the female predominance of thyroid disease in the general population [30], the effects of the block in males may not have been adequately measured. Third, as the block was established between the induction of general anesthesia and commencement of surgery, local anesthetic toxicity may have been masked. However, we encountered no fatalities.

## 5. Conclusions

PECS II may serve as a new pain relief modality, and may be a valuable addition to the current multimodal analgesic strategy for patients undergoing RATT. Although we have yet to investigate the optimal block duration and technique, regions of sensory loss, or possible complications, our preliminary study suggests that this new pain management technique reduces flap dissection pain and thus aids recovery. Appropriate analgesia during RATT remains challenging, but is key for postoperative recovery. We are prospectively validating the pain results to establish a pain control regimen for patients undergoing RATT.

## Figures and Tables

**Figure 1 cancers-14-04097-f001:**
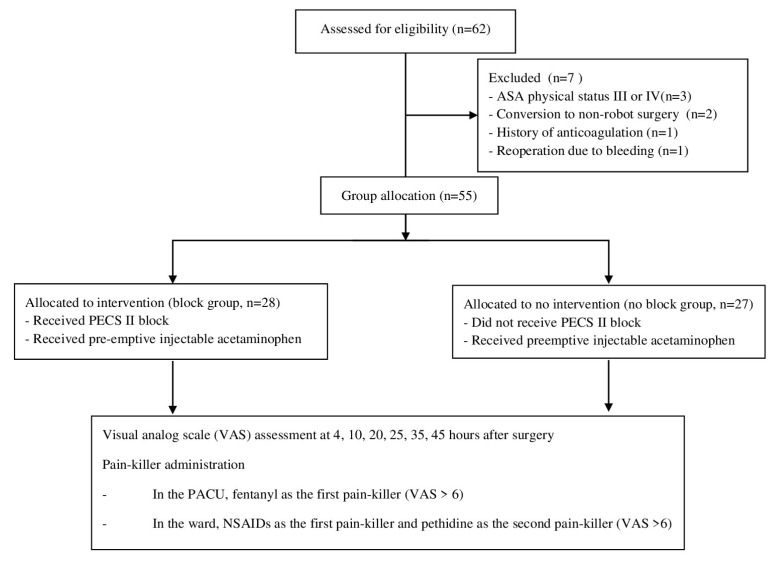
Study flow diagram.

**Figure 2 cancers-14-04097-f002:**
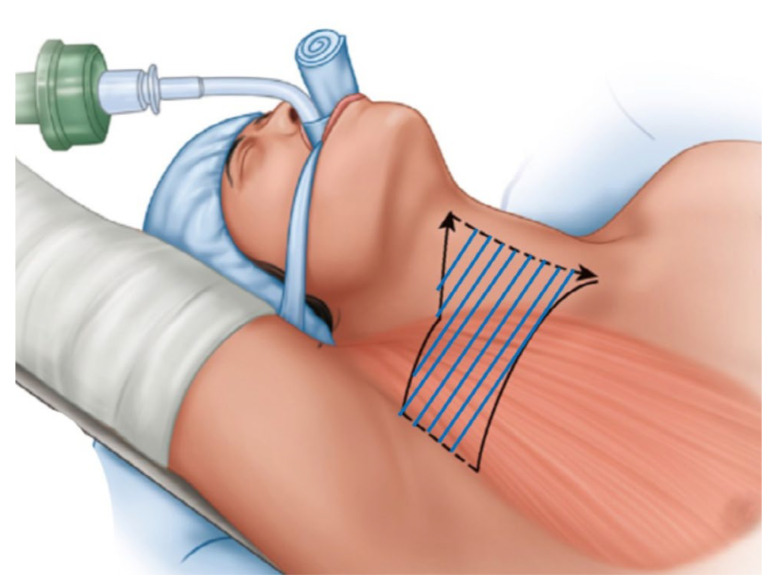
Extent of flap dissection (blue lines).

**Figure 3 cancers-14-04097-f003:**
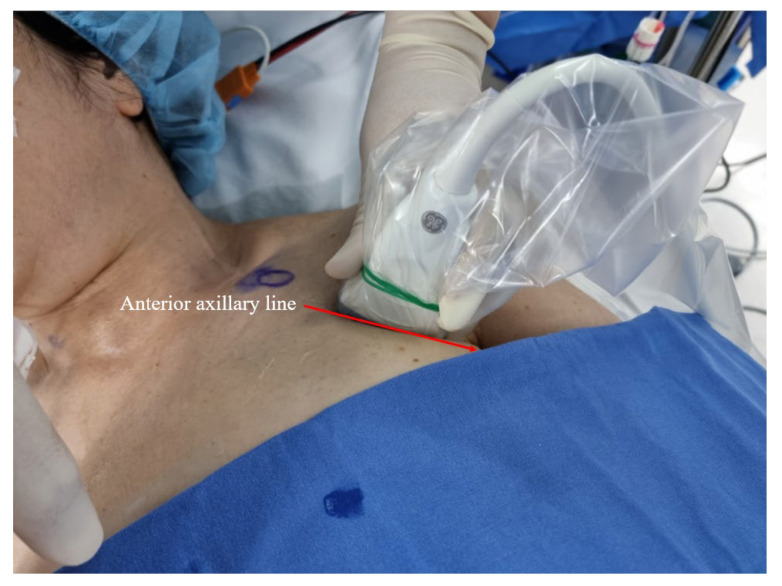
Advancement of the needle along the anterior axillary line.

**Figure 4 cancers-14-04097-f004:**
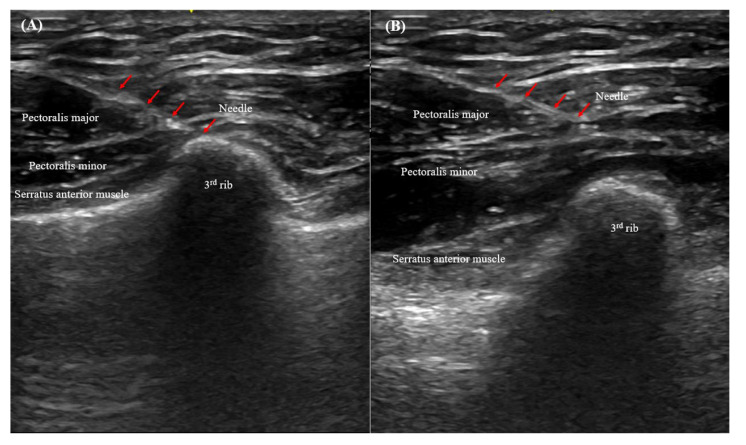
PECS II (**A**) between the pectoralis minor and serratus anterior muscles, and (**B**) between the pectoralis major and minor muscles.

**Figure 5 cancers-14-04097-f005:**
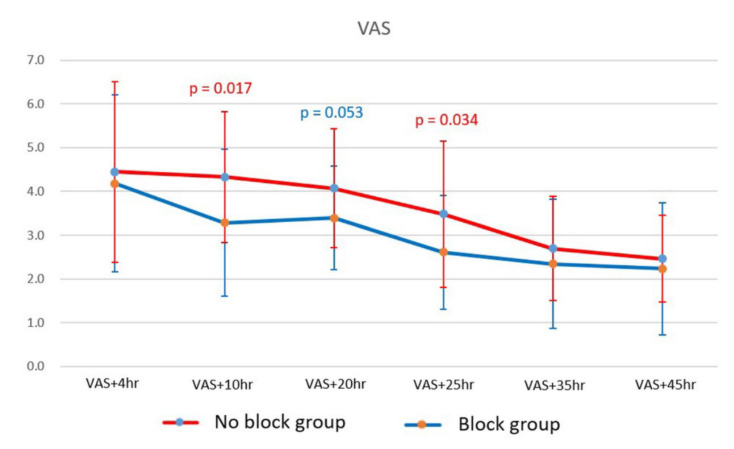
VAS scores of the two groups.

**Table 1 cancers-14-04097-t001:** Baseline clinicopathological characteristics.

Total 55 Patients	
Age (years)	41.5 ± 12.4 (range, 18–69)
Gender	
Male	3 (5.5%)
Female	52 (95.5%)
Extent of operation	
Lobectomy	48 (87.3%)
Total thyroidectomy	7 (12.7%)
Pathology	
PTC	45 (81.8%)
HCC	1 (1.8%)
NIFTP	3 (5.5%)
Follicular adenoma	4 (7.3%)
Graves’ disease	1 (1.8%)
Parathyroid adenoma	1 (1.8%)
Tumor size (cm)	1.1 ± 1.0 (range, 0.3–5.3)
Multifocality	19/49 (38.8%)
ETE	
No	19/49 (38.8%)
Minimal	30/49 (61.2%)
Thyroiditis	25 (45.5%)
Harvested LNs	6.2 ± 3.5
Positive LNs	1.0 ± 1.6
T stage	
T1/T2/T3a	44 (95.7%)/1 (2.2%)/1 (2.2%)
N stage	
N0/N1a	28 (60.9%)/18 (39.1%)
TNM stage	
Stage I/II	44 (95.7%)/2 (4.3%)

Data are expressed as the patient number (%) or mean ± standard deviation. Abbreviations: PTC, papillary thyroid carcinoma; HCC, Hurthle cell carcinoma; NIFTP, noninvasive follicular thyroid neoplasm with papillary-like nuclear features; ETE, extrathyroidal extension; LN, lymph node; T, tumor; N, node; M, metastasis.

**Table 2 cancers-14-04097-t002:** Baseline clinicopathological characteristics of the PECS II block and no-block groups.

	Block (n = 28)	No Block (n = 27)	*p*-Value
Age (years)	43.8 ± 11.7 (range, 24–64)	39.0 ± 12.8 (range, 18–69)	0.157
Female	28 (100%)	24 (88.9%)	0.111
Extent of operation			0.206
Lobectomy	26 (92.9%)	22 (81.5%)	
Total thyroidectomy	2 (7.1%)	22 (18.5%)	
Pathology			0.648
PTC	23 (82.1%)	22 (81.5%)	
HCC	0	1 (3.7%)	
NIFTP	2 (7.1%)	1 (3.7%)	
Follicular adenoma	2 (7.1%)	2 (7.4%)	
Graves’ disease	1 (3.6%)	0	
Parathyroid adenoma	0	1 (3.7%)	
Tumor size (cm)	0.9 ± 0.6	1.3 ± 1.2	0.160
Multiplicity	11/25 (44.0%)	8/24 (33.3%)	0.444
Minimal ETE	16/25 (64.0%)	14/24 (58.3%)	0.684
Thyroiditis	14 (50.0%)	11 (40.7%)	0.491
Harvested LNs	6.4 ± 3.1	5.9 ± 4.0	0.651
Positive LNs	1.0 ± 1.8	1.0 ± 1.6	1.000
T stage			0.352
T1	23 (100%)	21 (91.3%)	
T2	0	1 (3.7%)	
T3a	0	1 (3.7%)	
N stage			0.546
N0	15 (65.2%)	13 (56.5%)	
N1a	8 (34.80%)	10 (43.5%)	
TNM stage			1.000
Stage I	22 (95.7%)	22 (95.7%)	
Stage II	1 (4.3%)	1 (4.3%)	

Data are expressed as the patient number (%) or mean ± standard deviation. A statistically significant difference was defined as *p* < 0.05. Abbreviations: PTC, papillary thyroid carcinoma; HCC, Hurthle cell carcinoma; NIFTP, noninvasive follicular thyroid neoplasm with papillary-like nuclear features; ETE, extrathyroidal extension; LN, lymph node; T, tumor; N, node; M, metastasis.

**Table 3 cancers-14-04097-t003:** Postoperative VAS pain scores of the pectoral PECS II and no-block groups.

	Block (n = 28)	No Block (n = 27)	*p*-Value
VAS + 4 h	4.2 ± 2.0	4.4 ± 2.1	0.628
VAS + 10 h	3.3 ± 1.7 ^††^	4.3 ± 1.5	0.017
VAS + 20 h	3.4 ± 1.2	4.1 ± 1.4	0.053
VAS + 25 h	2.6 ± 1.3 ^††^	3.5 ± 1.7 ^††^	0.034
VAS + 35 h	2.4 ± 1.4 ^†††^	2.7 ± 1.2 ^†††^	0.350
VAS + 45 h	2.2 ± 1.5 ^††^	2.5 ± 1.0 ^†††^	0.511
No. of painkiller used			
In the PACU	0.7 ± 0.6	1.7 ± 0.5	<0.001
In the ward	1.0 ± 0.9	1.6 ± 1.6	0.062

Data are expressed as the mean ± standard deviation. A statistically significant difference was defined as *p* < 0.05. ^†^
*p* < 0.05, ^††^ *p* ≤ 0.01, ^†††^ *p* ≤ 0.001 compared to the VAS + 4 h in each group. Abbreviations: VAS, visual analog scale; PACU, post-anesthesia care unit. All patients used (1) Fentanyl as the first pain-killer in PACU, and (2) NSAIDs as the first pain-killer, and only three patients in the ward used Pethidine, an opioid analgesic.

## Data Availability

The data underlying this article cannot be shared publicly to maintain the privacy of individuals that participated in the study. The data will be shared upon reasonable request to the corresponding author.

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
