# Peer review of "Clinical Application of Pectoralis Nerve Block II for Flap Dissection-Related Pain Control after Robot-Assisted Transaxillary Thyroidectomy: A Preliminary Retrospective Cohort Study"

_cancers, 2022, doi:10.3390/cancers14174097_

Round 1

Reviewer 1 Report

The authors demonstrated that PECS II might be useful for pain control after RATT. Although I agreed the conclusions of this study, I have some concerns.

1) According to the authors, no study has examined the clinical utility of US-guided thoracic wall nerve block in RATT. This mention indicates that PECS II procedure for RATT may be investigational, which needs to get informed consents to the participants before study.

2) What was the patient selection criteria for PECS II? Some patients received PECS II block, while others did not. The results could be biased by the patient selection criteria.

Author Response

Point-by-point response letter

Reviewer #1

The authors demonstrated that PECS II might be useful for pain control after RATT. Although I agreed the conclusions of this study, I have some concerns.

Response) We sincerely appreciate your comments that make our work improve further.

1) According to the authors, no study has examined the clinical utility of US-guided thoracic wall nerve block in RATT. This mention indicates that PECS II procedure for RATT may be investigational, which needs to get informed consents to the participants before study.

Response) All pectoral nerve blocks were established by one highly experienced anesthesiologist (M.S.C.), in case of informed consent obtained from each patient a day before surgery (2.5 Ultrasonography-guided PECS II). An attached file (pdf) is a blank form of informed consent for the PECS II block.

2) What was the patient selection criteria for PECS II? Some patients received PECS II block, while others did not. The results could be biased by the patient selection criteria.

Response) Based on the order of the surgical schedule, all patients were classified into two groups: patients who received PECS II block (n = 28, 50.9%) and patients who did not receive the block (n = 27, 49.1%). The patients with an odd-numbered position in the surgical schedule were assigned to the block group, while those with an even-numbered position were assigned to the no block group (2.2 study population). As it was preliminary and retrospective, selection bias may have been present. We are prospectively validating the pain results to establish a pain control regimen for patients undergoing RATT.

We thank you and the reviewers for the insightful comments. We believe that our manuscript has been improved as a direct result of the review process. We hope that the revised manuscript is now suitable for publication in CANCERS.

Reviewer 2 Report

You studied pain relief in patients with TA. I appreciate your research. If this process is standardized, it will result in good treatment for TA patients. I consider your research to be of high value.

- Why is the percentage of women so high?

- The position of the line in table 2 is incorrect. (lining)

- Is it necessary to correct the contents of table 3?

No. of painkiller used

In the PACU

- Do you routinely perform nerve block procedures?

- How much time does the nerve block process take, and is it performed by one person? I wonder if it is a part that can be applied universally to everyone.

Author Response

Point-by-point response letter

Reviewer #2

You studied pain relief in patients with TA. I appreciate your research. If this process is standardized, it will result in good treatment for TA patients. I consider your research to be of high value.

Response) We sincerely appreciate your comments that make our work improve further.

- Why is the percentage of women so high?

Response) Because of the role of sex hormones, a female predominance is well recognized for most of thyroid diseases, ranging from thyroid dysfunctions to some form of congenital thyroid dysgenesis, or diffuse or focal nodular enlargement of the thyroid tissue, including thyroid cancer. Those thyroid disease features may contribute to sex proportion in our study, and future studies should consider sex distribution originating from the diseases (limitation).

- The position of the line in table 2 is incorrect. (lining)

 Response) Per your comments, we revised the alignment in table 2.

- Is it necessary to correct the contents of table 3?

No. of painkiller used

In the PACU

 Response) Per your comments, we revised to ‘In terms of painkiller usage, the block group required fewer rescue analgesics in the PACU than the no-block group, although the difference was not statistically significant in the ward’ (Abstract), (3.2. Postoperative pain in the block and no-block groups).

- Do you routinely perform nerve block procedures?

 Response) Yes, we have regularly provided regional analgesic service (such as, PECS II) to patients undergoing surgeries, including RATT.

Current pain control strategies are based on multimodal approaches, such as the enhanced recovery after surgery (ERAS) protocol; interventional locoregional anesthesia are an essential component of multimodal analgesia in this regimen. Their use enhances analgesia and reduces or eliminates the use of opioids in the early postoperative period. As the ERAS protocols continue to evolve, so do interventional locoregional anesthesia techniques. Some traditional regional methods are being substituted by more selective approaches to minimize the unwanted effects, such as hemodynamic changes or motor block, which may delay early rehabilitation and recovery. A number of new ultrasound-guided fascial plane injection techniques, distal nerve blocks, and selective periarticular injection have been described with the aim to find a balance between efficacy, simplicity, safety, and sensory-motor block ratio. However, many newer techniques have not been standardized, and the paucity of data on their efficacy makes their implementation into pragmatic protocols difficult.

Therefore, our study has strengths that PECS II may provide early pain relief; patients who received this block recovered more comfortably and require fewer painkillers in RATT. Although further validation is still required, PECS II may be a valuable analgesic option for reducing flap dissection pain and stress, as well as opioid consumption, during early recovery after RATT. In RATT settings, the routine block, including PECS II, may be one of major components for establishing the multimodal ERAS protocol.

- How much time does the nerve block process take, and is it performed by one person? I wonder if it is a part that can be applied universally to everyone.

Response) During preliminary study period, all PECS II blocks were performed by one attending anesthesiologist (M.S.C.), who specialized for regional analgesia, to keep the block quality high. Because of our retrospective nature, unfortunately we did not measure the time consumption of PECS II block, but usually not too much time is required that, in one example (we measured his time consumption of PECS II block), total spending time is approximately within 5 min from sterile drape to finishing the local anesthetics injection without any complications. In block time consumption related literatures, little studies have been reported because of various confound factors existed, such as surgery types, block techniques and sites, patients’ anatomical variations, and learning curves for the individual block.

 In the clinical implication of PECS II block, the block can be useful in delivering regional analgesia for a wide variety of surgical procedures including insertion of breast expanders and submuscular prostheses, ports, pacemakers, implantable cardiac defibrillators, anterior thoracotomies, anterior shoulder surgery, tumor resection, mastectomies, sentinel node biopsy, and axillary dissection. In the contraindications, patients refusal or infection at the site of local anesthetic injection are ‘absolute’ contraindications to performing a PECS II block (including PECS I block – sister block). Anticoagulation may be a ‘relative’ contraindication to PECS II block, although there are no specific guidelines. The 2018 ASRA consensus (DOI:10.1097/AAP.0000000000000700) statement does not specifically address PECS blocks and anticoagulation. In the complications, there are rare with the use of ultrasound guidance, as the pleura and major blood vessels are visible throughout the procedure. The most common complications are pneumothorax, infection, local anesthetic toxicity/allergy, vascular puncture, and failed block. Therefore, although the specific guideline or consensus is nor suggested fully, the clinicians are mandatory to meticulously perform the block under multiple monitoring, (if he/she is a novice, supervision is unconditionally required) and early identify the vulnerable patients with poor ultrasound images, anticoagulation history, and variant anatomical structures.

We thank you and the reviewers for the insightful comments. We believe that our manuscript has been improved as a direct result of the review process. We hope that the revised manuscript is now suitable for publication in CANCERS.

Round 2

Reviewer 1 Report

Thanks for your response, but I still have ethical concerns about the informed consent.

The allocation of patients according to their participation orders indicated that the design of this study were prospective, non-randomized trial. As I mentioned earlier, you must get informed consents from the participant about the purpose of this trial. However, the informed consent form that you attached is the consent for PECS II procedure, not for this investigational or experimental study.

Therefore, in my opinion, the authors should obtain the institutional review board approval again for the prospective study design, and get informed consents to the participants for this trial (not consents for PECS II block).

Author Response

Response) Thank you for your comments, and we totally agree your concerns regarding the study design. The present study has a limitation in terms of a preliminary results with retrospective nature (page 24), but our study strength is the first report in the clinical application of PECS II block for flap dissection-related pain control after RATT and provides the potentially analgesic role of PECS II block for patients undergoing RATT, including axially flap dissection procedure. As we mentioned in the discussion (pages 23 and 24), PECS II block has clinically applied for a feasible and practical analgesic treatment in patients undergoing upper-trunk area surgeries, such as breast surgery, and the evidence in availability and safety of PECS II block has widely shared in the clinical community. Because of retrospective nature, regretfully, we are not available to obtain the informed consent for the study, but we fully explained the effect and complications of PECS II block to our patients, and regularly and legally obtained the informed consents for the PECS II block. Our IRB approved this retrospective study under robust and thorough review. Based on our hypothesis and results (PECS II block may be a valuable analgesic option for reducing flap dissection pain and stress, as well as opioid consumption, during early recovery after RATT. PECS II may provide early pain relief; patients who received this block recovered more comfortably and require fewer painkillers), we have performed the prospective, randomized, double-blinded trial (with obtaining new IRB approval and patient informed consent) that validates the pain results to establish a pain control regimen for patients undergoing RATT. Therefore, sharing our preliminary retrospective results may activate the serially relevant studies and increase the interest in the analgesic role of PECS II block for patients suffering thyroid cancers undergoing RATT.        

We thank you and the reviewers for the insightful comments. We believe that our manuscript has been improved as a direct result of the review process. We hope that the revised manuscript is now suitable for publication in CANCERS.

Sincerely,

Kwangsoon Kim, MD, PhD

Round 3

Reviewer 1 Report

I agreed to accept this manuscript.